# Incarcerated individuals and education programmes in Nigeria: A task for social workers

**Ijeoma B. Uche**[1☯], **Okala A. Uche**[1☯], **Uzoma O. Okoye**[1], **Blessing Onyinye Ukoha-Kalu**[2,3]*

**1** Department of Social Work, University of Nigeria, Nsukka, Enugu State, Nigeria, **2** Department of Clinical Pharmacy and Pharmacy Management, Faculty of Pharmaceutical Sciences, University of Nigeria, Nsukka, Enugu State, Nigeria, **3** School of Medicine, University of Nottingham, Nottingham, England, United Kingdom

☯ These authors contributed equally to this work.
* blessing.ukoha-kalu@nottingham.ac.uk

**Data Availability Statement:** All relevant data are within the paper.

**Funding:** The author(s) received no specific funding for this work.

## Abstract

Much is not known about education programmes for Nigerian incarcerated individuals. Consequently, different correctional institutions worldwide have different forms of correctional education offered to incarcerated individuals. Nigerian incarcerated individuals perceive that there are implementations of education programmes offered to them. However, little or nothing is known about how incarcerated individuals perceive these education programmes. To this end, this study ascertains the incarcerated individuals' perception of education programmes as well as the expectations of social work profession in ensuring that incarcerated individuals in correctional institutions are provided with quality education. In-depth interviews involving 20 convicted incarcerated individuals from Owerri correctional centre were conducted. Thematic analysis was used in analyzing data generated for the study. Findings show that education programmes are available and essential, but the quality of the programme does not go down well with the incarcerated individuals. It also revealed that incarcerated individuals were not allowed to decide on the type of education programme to be involved in. To this end, their participation in the programme is jeopardized. The findings further revealed that little or nothing is known about social workers in prison education. Therefore, the inclusion of correctional social workers as an integral part of education programmes that aim to reform, rehabilitate and reintegrate incarcerated individuals becomes necessary.

## Introduction

Prison education, which is also called correctional education has been defined by many researchers as the education designed for prisoners to equip them with veritable skills and knowledge to enable them to be productive on release [1–3]. These Correctional education programmes for incarcerated individuals include carpentry, welding, adult literacy, tailoring, self-development (stress/anger management), violence (violence/aggressiveness management), addiction (substance abuse, drug/alcohol addiction), education, job skills, leisure, art, and

**Competing interests:** The authors have declared that no competing interests exist.

spirituality [4, 5]. Correctional education is one of the rehabilitations programs provided for incarcerated individuals. Social workers in prison setting seek to access quality rehabilitation programmes for incarcerated individuals; run educational programmes one-on-one or in groups; identify appropriate educational and vocational opportunities that help incarcerated individuals with transitioning back into their communities [6]. Involvement in a prison education programme is a step forward in demonstrating a commitment on the part of social work to serve vulnerable and often neglected populations [7].

Several reports have shown the benefits/impacts of prison education. 67.8% of incarcerated individuals who received education programmes while in prison get jobs easier on release [8]. Education makes incarcerated individuals busy and active, preventing idleness and misbehavior opportunities. It also increases human resources, improves overall psychological well-being, and provides specific skills at the same time [9]. [10] stated that education in prison is a prevailing tool in the offender's reformation, rehabilitation, and self-development. Education of prison-incarcerated individuals could be seen to hold the prison them busy. Participation in correctional education, while incarcerated, has been reported to be positively associated with lower recidivism risks [11–16] and higher employment chances [11, 17].

Despite the benefits/impacts of education on incarcerated individuals and efforts of some international organizations to promote correctional education, there still exists a high rate of recidivism globally [18–21] which is attributed to low or no participation in education programmes while in incarceration [22, 23]. Extant research has shown that not all correctional education results in effective rehabilitation [16, 24]. In sub-Saharan Africa, which has high rates of incarcerated individuals, only fewer percentage of them participate in education programmes. In Luzira Prison, Uganda, 10% of incarcerated persons are enrolled in prison education [25]. In Enugu prison, out of a total of 2,466 incarcerated individuals, only 129 sat for National Examinations Council Nov/Dec 2018 examination [26]. Also, 2 incarcerated individuals are currently running their PhD programmes with the National Open University of Nigeria [27]. [28] reported that most prisons in Zambia lacked counsellors and psychologists to attend to incarcerated individuals. In the same vein, there is a lack/shortage of caregivers such as social workers in Nigerian prisons. Nigerian prisons are overcrowded with awaiting trial incarcerated individuals. This category of incarcerated individuals does not have access to education programmes like the convicted individuals [4].

Certain constraints have been associated with prison education, in developed and developing countries. Transfer of incarcerated individuals from one prison to another prison, age, length of service, lack of funding, lack of access to an optimal learning and study environment, stigma, overcrowding, poor management and lack of access to internet facilities are challenges that affect prison education [29–31]. However, the goals of Education for All (EFA) in 2015 are yet to be acknowledged because education policies are allegedly poorly enforced in Nigerian prisons [32]. Assisting prisoners to gain knowledge, skills, and competencies is an integral aspect of rehabilitation and reintegration into society [33].

According to [34] no meaningful sustainable development can take place without the involvement of professionals. Professionals such as social workers must collaborate with correctional institutions in enhancing educational programmes. Owing to the deplorable conditions of our correctional centres in Nigeria, the need for social education becomes paramount. This form of education not only aims at eliminating illiteracy, it also has the major role of accelerating the flow of technical knowledge. Moreover, it motivates correctional centres to adopt and maintain useful practices in health and nutrition among others.

The social worker assists the offender by providing knowledge and skills of the profession in a corrective manner [35]. Social workers have the major role of raising awareness and consciousness of incarcerated individuals on the meaningful adjustment to the prison condition.

They play very important roles in the rehabilitation of offenders, delivering both pre-release education programmes and group discharge programmes [36, 37]. The main concern of the social worker is to make incarcerated individuals understand positive change, a certain quality of life, and/or protection from harming others [38]. Social workers provide counselling for incarcerated individuals; they link prison staff who teach incarcerated individuals life skills in rehabilitation to resources that will develop their capacity [39]. However, Nigerian correctional centres are yet to come to terms with the tasks of social workers in correctional settings. This is in tandem with the views of [40] that social work has greatly abandoned the field of corrections.

Several studies on prison education measure the impact of correctional education programmes on recidivism and employment rates among ex-prisoners [16, 41]. Nevertheless, little is known about how incarcerated individuals perceive such programmes while in custody [42]. How incarcerated individuals perceive education programmes differ from one prison to the other and even within the same prison and there is a need for social workers also known as correctional treatment specialists [35] in assisting incarcerated individuals through correctional education to reintegrate back into society upon release. This, therefore, necessitates the need for specific data from different prison settings. This study investigates incarcerated individuals and education programmes in Nigeria: A task for social workers. The study addressed the following research questions: (1) What are the education programmes available for incarcerated individuals? (2) What are the views of incarcerated individuals and essence of correctional education? (3) What is the quality of correctional education programmes and incarcerated individuals' participation? (4) What are the views of incarcerated individuals on social workers and correctional education? Addressing these research questions will enhance the participation of incarcerated individuals in correctional education within the study area and possibly other correctional institutions in Nigeria.

## Methods

### Study design, area, and population

This study used a phenomenological approach that allowed incarcerated individuals to describe their experiences and thoughts about correctional education. The approach allowed the researchers to report from the perspectives of those with direct lived experience [43]. Qualitative studies provide an opportunity for exploring the diverse experiences and opinions of individuals [44, 45]. Our choice of the qualitative inductive approach was to make for flexibility in the choice of methods and to enable the study findings to emanate naturally from thematic data [45, 46].

The study was in Owerri prison, Imo state, Nigeria. The prison is a maximum-security centre which holds both male and female incarcerated individuals and has the original capacity of carrying 548 incarcerated individuals. At the time of data collection, Owerri prison had a total population of 2,372 incarcerated individuals (2,278 males and 94 females) (Record office Owerri prison, 2021). Out of the total population of 2372, only 183 are convicted individuals (181 males and 2 females), while the remaining 2,189 were awaiting trials. It offers various educational programmes for incarcerated individuals. Undoubtedly, these education programmes are meant to help incarcerated individuals acquire skills and be gainfully employed upon discharge from prison; social workers are needed if this aim should be actualized. The centre was chosen because it is central and easy for researchers to access it. Besides, two of the researchers are from the area where the prison is located. Furthermore, the fact that the centre accommodates both male and female incarcerated individuals gave it an added advantage.

## Sampling procedure and participants

The purposive and availability sampling techniques were used in the study. It is purposive in the sense that education programmes in Nigerian prisons are only made for convicted individuals. This category of individuals, therefore, qualified for the present study. 183 convicted individuals that were available met the condition of being selected.

The officer in charge gave us a list of the convicted individuals (183) and we looked through their years of admission to ascertain the older individuals (3 years of service and above). He (the Officer in charge) assisted in identifying this category of individuals, we explained the objectives of the study, the benefits and the risks associated with the study as well as assuring them of their anonymity and confidentiality. This was not an interventional study so individuals who did not participate in the study received the educational programme as those who participated in the study. Out of a total of 183 convicted individuals, 20 incarcerated individuals met up with this criteria. This gave every convicted individuals that met the condition an equal chance of being selected and the probability of selecting a participant was not affected by the selection of other participants in the population [46, 47]. The justification for our choice of 20 participants for the interviews was that since we are conducting a small-scale qualitative study, that sample size allowed for saturation and manageable data [48, 49]. According to [50], attaining data saturation has nothing to do with quantity but the richness of data.

## Data collection

In-depth interviews (IDIs) served as the instrument for data collection. It was used to elicit information from the participants. The interviews were conducted one after the other at the prison Chapel. The lead researcher moderated the interview, while other researchers served as recorders and note-takers. The IDI guide consists of four questions with probes designed by researchers to explore incarcerated individuals' perceptions of prison education. The interviews were conducted in the English language. However, three incarcerated individuals opted for the Igbo language which we consented to. Data collected were analysed in the English language. All the researchers are well-grounded in the Igbo language. The field notes were compared with transcribed contents to ensure coherence and accuracy. Participants were briefed about the objectives of the study and assured of all ethical considerations concerning their participation such as confidentiality, anonymity, and freedom to withdraw at any point in time before engaging them in the interviews. The participants gave both oral and written consent prior to the interviews. Pseudonyms were used throughout the article to protect the identities of the participants. The interviews were audio-recorded to enable the researchers/interviewers to focus on the participants and their non-verbal cues. Each interview lasted for 30 to 35 minutes. A pilot study was conducted at Nsukka prison. Five respondents were used in the pilot study. This was to ensure that the questions were appropriate in obtaining reliable information on the phenomenon under investigation.

## Data analysis

The transcribed data were analyzed using Nvivo 12 software. The researchers conducted multiple reads of both separate transcripts and the entire interview, comparing them with the Nvivo coding to reflect on participants' narratives, and ensure the validity and reliability of results. We engaged in thorough and attentive multiple readings of the transcripts until we identified emerging codes and patterns, which we then organized into themes and used as quotes while presenting our findings. The qualitative data generated from the IDIs including the non-verbal communications of the respondents were recorded, edited, and interpreted. The edited and interpreted responses were all analyzed qualitatively. The data analysis focused on the research

questions. These research questions formed the themes for data analysis. They include: (1) What are the education programmes available for incarcerated individuals? (Education programmes available for incarcerated individuals) (2) What are the views of incarcerated individuals and essence of correctional education? (Incarcerated individuals' perception and essence of correctional education) (3) What is the quality of correctional education programmes and incarcerated individuals' participation? (The quality of correctional education programmes and incarcerated individuals' participation) (4) What are the views of incarcerated individuals on social workers and correctional education? (Knowledge of social workers and correctional education).

### Ethical consideration

Ethical permission to conduct this research was sought from the Strategic Contacts, Ethics and Publications (STRACEP) at the University of Nigeria, Nsukka (STRACEP–UNNEC/05/0021/10-ST03/0024). Written consent was obtained from each participant before enrolling in the study.

## Results

The profile of the participants is described in Table 1. Four themes were generated from the study: (i) Education programmes available for incarcerated individuals (ii) Incarcerated individuals' perception and essence of education programmes (iii) The quality of correctional education programmes and incarcerated individuals' participation (iv) Knowledge of social workers and correctional education.

### Theme one: Education programmes available for incarcerated individuals

The participants revealed that education programmes designed for the incarcerated individuals in Owerri prison were basically categorized into five. They include Vocational and Technical Training (VTT), Adult Basic Education (ABE), Secondary School Equivalence (SSE), Tertiary Education (TE), and Computer training. Tertiary Education is being organized by the National Open University of Nigeria (NOUN) for incarcerated individuals, while prison officials and some non-governmental organisations co-ordinate the other education programmes for the incarcerated individuals. The transcript analysis revealed that there is an availability of correctional education programmes for incarcerated individuals. One of the participants, Alfred, aged 25 and was sentenced to 5 years said, "*Yes, we have some education programmes in this centre. There is vocational and technical education, adult education and senior secondary exam for those preparing for West African Examinations*". Another participant, Jerry with 10 years prison sentence and 38 years noted, "*. . .NOUN co-ordinate higher education for incarcerated individuals and even enrolled some of us in skill acquisition programmes*". A remarkable assertion was made by Castro who was 48 years and sentenced to 8 years in prison, "*Hmmm. . .. there is computer training, not ICT. It is computer training because the few desktop computers we have here are used to train incarcerated individuals to learn typing skills*".

### Theme two: Incarcerated individuals' perception and essence of correctional education

The participants were unanimous in their perception of the existence and essence of education programmes available in correctional centres. Their perceptions were based on the usefulness of correctional education. All the participants perceived correctional education as very essential for preparing them for life after the "wall". Their perception of the essence of prison

**Table 1. Profile of participants.**

| S/N | Pseudonyms | Age | Prison sentence (years) | Period served (years/months) | Occupation before conviction | Education status before conviction | Marital status | Gender |
|---|---|---|---|---|---|---|---|---|
| 1. | Nelly | 18 | 4 | 3.4 | Student | Senior secondary education | Single | Female |
| 2. | Alfred | 25 | 5 | 3.0 | Farmer | Junior secondary education | Single | Male |
| 3. | Sam | 24 | 7 | 4.4 | Civil servant | Tertiary education | Married | Male |
| 4. | Elly | 33 | 15 | 3.2 | Civil servant | Tertiary education | Married | Male |
| 5. | Armstrong | 45 | 10 | 4.8 | Trader | Senior secondary education | Married | Male |
| 6. | Jerry | 38 | 10 | 6.1 | Student | Tertiary education | Married | Male |
| 7. | Mirabel | 46 | 12 | 3.0 | Trader | Tertiary education | Widowed | Female |
| 8. | Fetch | 31 | 6 | 3.2 | Civil servant | Tertiary education | Married | Male |
| 9. | Bon | 25 | 10 | 3.1 | Farmer | No formal education | Single | Male |
| 10. | Gab | 21 | 15 | 5.5 | Trader | Senior secondary education | Single | Male |
| 11. | Joel | 44 | 14 | 7.6 | Civil servant | Tertiary education | Married | Male |
| 12. | Dan | 49 | 12 | 5.6 | Civil servant | Tertiary education | Married | Male |
| 13. | Castro | 48 | 8 | 3.3 | Civil servant | Tertiary education | Divorced | Male |
| 14. | Andy | 30 | 5 | 3.1 | Artisan | Primary education | Separated | Male |
| 15. | Zanny | 37 | 7 | 5.6 | Civil servant | Tertiary education | Divorced | Male |
| 16. | Ben | 26 | 10 | 4.6 | Farmer | Tertiary education | Single | Male |
| 17. | Dom | 19 | 8 | 3.7 | Unemployed | Senior secondary education | Single | Male |
| 18. | Mast | 29 | 6 | 3.3 | Trader | Tertiary education | Single | Male |
| 19. | Zitto | 20 | 8 | 3.9 | Civil servant | Tertiary education | Single | Male |
| 20. | Namdy | 34 | 14 | 4.9 | Civil servant | Tertiary education | Single | Male |

**Source**: Field survey, 2021

education has both instrumental and existential values. For many of them, the education programme is essential for their reintegration into society, they acquire new skills, it gave them opportunities to learn, provides incarcerated individuals with confidence and enabled them to have a positive outlook in life. Some of the participants perceived correctional education as a means of making money which they sent to their individual families. Others said that it had improved their mental well-being because of the teamwork involved in it. One of the participants, Ben, aged 26 and was sentenced to 10 years said, "*Staying in prison is bitter because of restrictions but education has made it to be a little bit sweet because we are gaining some skills that will be useful to us and society*". Another participant Bon with 10 years prison sentence and 25 years added that "*. . .education has given me a second chance in life. I learnt some skills that I did not have before coming here*". Another participant, Jerry, aged 38 also said, "*Education in prison is encouraging because of the teamwork involved in it. I could associate with fellow incarcerated individuals and make friends easily*".

The expression of another incarcerated individual was captured in the quote:

*Indeed, my stay here is not in vain. I am serving a jail term and at the same time benefit from it. The education provided here impacted meaningfully on my behavior and by the time I complete my sentence, I will be a better citizen. My meaningful engagement in educational activities has made me focus on the positive outlook of life* [Fetch: Age 31years, sentenced 6 years].

The analysis of the transcript also revealed that correctional education has given incarcerated individuals the opportunity to make a living. Some of the participants stated that they

have been able to make money consequent upon the skills they have learnt through correctional education. Andy who was sentenced to 5 years in prison and aged 30 said, "*I can make money that I couldn't make before here. I sew uniforms for some staff and they pay me afterwards and I normally send the money to my people to help in their financial problem*". Another participant Armstrong, aged 45 and was serving 10 years in prison revealed, "*The good thing they did for us here is the provision of education for incarcerated individuals. If not for keeping us busy with some skills, there would have been crime inside crime*". Zitto who was serving 8 years imprisonment and aged 20 noted, "*With education, the prison harsh condition is more bearable, and it gives one hope of another chance in life*". In his words, Mast, aged 29 and has 6 years prison sentence said, "*Prison education is very useful to me. It has boosted my self-esteem since I can comfortably boast of some skills*".

Another participant's expression was captured thus:

*Government and prison authorities did well by incorporating education as part of the programmes for incarcerated individuals. Education has the ability to change that person who was a criminal or offender into a law-abiding citizen. It should be made compulsory as one of the criteria for incarcerated individual's release upon completion of jail term* [Sam; 24 years, 7 years sentence].

From the above accounts, it was obvious that participants understood prison education and its benefits. Howbeit, for some of the university-educated incarcerated individuals who were civil servants before incarceration, prison education may not actually yield the expected positive results in terms of getting employment when released as ex-incarcerated individuals find it difficult to get a job when they are done with prison service. Hear them: According to Dan who was 49 years and was serving 12 years in prison said, "*You won't get any job once it is ascertained that you were once a prisoner*". Joel, who was 44 years and serving 14 years jail term said, "*. . .the skills acquired through education can help the ex-incarcerated individual to be self-employed but not getting a white-collar job. There is a stigma attached to that name ex-prisoner*". Another participant, Namdy, aged 34 years and was sentenced to 14 years in prison noted, "*It is not always true that any incarcerated individuals who acquired education while in prison will turn out to be a law-abiding citizen. There are exceptions to the rule*".

Furthermore, some participants expressed doubt about correctional education. These expressions were captured in quotes:

*It is true that I am aspiring to better my life upon discharge from prison. However, I am also aware that even the government that made education possible for incarcerated individuals will not even employ me in any of their establishments. No organization including the Nigerian government would like to employ somebody with a criminal record* [Zanny; Age: 37; sentenced to 7 years in prison].

*There are limits to the benefits of prison education. I was meaningfully employed before my conviction. However, I am not too sure about being re-absorbed at the end of my service because of the stigma attached to being an ex-convict* [Elly; Age: 33; 15 years sentence].

*It is always very easy to say that prison education helps incarcerated individuals to get jobs upon release. I know little or nothing about ex-convicts who were employed when they are done with their stay in prisons. Usually, after a background check and it is ascertained that the applicant has once been in prison, the ex-convict has a very slim chance of being employed* [Castro; Age: 48; 8 years' sentence].

### Theme three: The quality of correctional education programmes and incarcerated individuals' participation

Although there was an immense acknowledgement of the vital roles of correctional education in preparing incarcerated individuals for life outside the prison, our findings revealed another important aspect of prison education that has to do with the quality of education and participation of incarcerated individuals in these programmes. Some of the respondents revealed that the types of education programmes do not encourage active participation in the programme. One of the participants Dom, aged 19 and was sentenced to 8 years said, "*There are security issues presently in prisons; we are in constant fear of being attacked. This to a large extent has affected our participation in education programmes after all the authority is not even serious about that*". "*Learning tools are not adequately provided in this centre. It makes the whole exercise boring and demoralizing*" [Ben]. A female voice, Mirabel, 46 years and sentenced to 12 years in prison said, "*Just very few incarcerated individuals have access to Senior Secondary School Examination. I and some other incarcerated individuals would have loved to further our education but were not given the opportunity because of finance*".

The expressions of other participants were captured thus:

*Nigerian correctional centres and prisons in most African countries have not attained that level of making correctional education enjoyable and encouraging. I do not participate in any of the programmes because Information and Communication Technology (ICT) is not part of the education programme here. Imagine in this internet era one is denied access to the internet simply because one is a prisoner* [Castro; Age 48; 8 years sentence].

*I stopped participating in education because of the language barrier. The instructors are fond of using words I found difficult to understand. Some of the instructors use ambiguous words in educating us. I feel inferior whenever such ambiguous words are used. The use of ambiguous words is my problem* [Alfred; Age 25; 5 years sentence].

*The quality of the education programme is of a low standard. There are irregularities on the side of education instructors. Some of them skip classes as a result they do not care because they are overworked and underpaid. This is a big barrier to participating in education programme* [Dom; Age 19; 8 years sentence].

*Transferring incarcerated individuals or releasing them affects participation in education. A transferred or released prisoner automatically quits the programme he/she was engaged in. In the case of transfer, the incarcerated individual may not have the opportunity to continue with the exact programme he participated the previous prison if that programme is not available in the current prison* [Armstrong; Age 45; 10 years sentence].

*There are very limited education programmes in this centre. As a result of this incarcerated individuals are restricted to only a few available ones. We are not allowed to decide which programme to participate in. Incarcerated individuals should be allowed to take a decision on the programme suitable for him/her* [Zitto; Age 20; 8 years sentence].

However, respondents were of the view that participation in rehabilitation programmes such as education programmes was very low as most of the incarcerated individuals do not show much interest in being involved in the programmes. For them, it was a waste of time and energy. Hear them:

*I can assure you that out of 13 incarcerated individuals in my Cell; only three of us are interested in education programme. The rest sometimes make a mockery of us insisting that we are disturbing ourselves by learning what we will never practice* [Alfred].

Another said participant Zanny who was 37 years and sentenced to 7 years said, "*In this prison, very little percentage (5%) of convicted individuals have access to education programmes, the rest are awaiting trial persons who have no access to education programmes and it affects participation*".

## Theme four: Knowledge of social workers and correctional education

Transcript analysis further revealed that participants have little or no knowledge of social workers in promoting correctional education. Many of the participants do not know who the social workers are and what they can do for the incarcerated individuals. Fetch who was sentenced to 6 years in prison and aged 31 said, "*Are they those religious people that normally come here on Sundays*?" 44 years old Joel with 14 years prison sentence said, "*Many people visit us here and I have not heard any of them mention social worker*". Additionally, Mirabel who was sentenced to 12 years in prison and aged 46 remarked, "*Obviously, I do not know the social workers and I also do not know the services they offer. I know that charitable organizations do visit us*". "*I don't know who they are but if they are the ones to liaise with the government in making correctional education better, please call them*" [Zanny].

However, two out of the twenty participants claimed they are aware of social workers. Gab who was sentenced 15 years and aged 21 noted, "*I saw them in the hospital when I was sick. They were working hand in hand with the doctors, but I don't know what they will do for us here*".

*Social workers exist in organized societies where they are employed to work in different places including prisons. They counsel, advocate and mediate on behalf of the incarcerated individuals. I have read much about them, they also help in teaching the incarcerated individuals* [Castro].

## Discussion

The findings presented in the study came from 20 convicted incarcerated individuals in Owerri correctional centre, Imo State, Nigeria who expressed their views on correctional education. The findings show that incarcerated individuals perceived the existence of correctional education and its usefulness. The limited knowledge of social workers and their roles in the study area shows that there is a need for social workers in promoting education programmes in prisons. Social workers help to rehabilitate incarcerated individuals, counsel, teach life lessons and advocate for better educational policies for prison-incarcerated individuals [35, 51, 52]. This is synonymous with findings from other studies [25, 37].

Despite the usefulness of correctional education, the study findings revealed that participants are skeptical of what education can do to incarcerated individuals upon release in terms of finding a job. Participants were of the view that after acquiring the sought education while in prison, getting a meaningful job has always been difficult as nobody wants to associate with somebody who has a criminal record. This proves the fact that there are limits to the essence of correctional education thus making it ineffective [16, 24]. The finding revealed that correctional education does not always reform incarcerated individuals' behaviour [53, 54]. It contradicts the findings of [11] that incarcerated individuals who engage in education have an increased chance of finding work after their release from prison. This contradiction may be associated with the limitations or challenges of correctional education programmes.

Very vital to the findings of this study is the quality of correctional education and participation in the programme by incarcerated individuals. Participants reported that the nature of

education in prisons affected their participation in the programme. They see correctional education as of low quality since it is faced with challenges that affect its effectiveness. These include a lack of adequate tools, the use of ambiguous English words that are difficult for the participants to understand, the release or transfer of incarcerated individuals from one prison to the other and the prison environment among others [29, 55]. Also, it was further revealed that participants were never asked to make their choice in terms of choosing the type of education they want to participate in. They also view the extent of participation from incarcerated individuals as very low [56]. Correctional education is to some extent available in Nigerian prisons, but the quality varies from what is obtainable in the western world. All countries provide education for incarcerated individuals inside the prison, but the quality and type vary [57].

Our results further revealed that most of the participants were not aware of social workers in prison. There was an overwhelming acceptance of the lack of social workers in correctional institutions. Only two of the participants have little knowledge of social workers. In lieu of this, there is a need for professional support as posited by [58, 59].

## Research implications

Most Nigerian correctional institutions are yet to be fully aware of social workers and their roles in the education of incarcerated individuals probably because the profession is still new. There is a need for social workers to train more professionals who will help in creating awareness in the correctional institutions of the services they offer. Rehabilitation is the key focal point of correctional social workers. Correctional social workers counsel, teach life lessons, lead rehabilitation groups, and help incarcerated individuals plan to reintegrate after they are released back into the community. Social workers need to have excellent skills that help them communicate with incarcerated individuals in a caring and competent way [51]. Correctional social workers may be suitable in advocating for better education policies for incarcerated individuals as indicated in the study findings. Correctional social workers in Nigerian correctional centres and particularly in Owerri prison are yet to fully activate wider activism that enhances education programmes in the prison.

There is a need for the inclusion of social workers as an integral part of education programmes that aim to reform, rehabilitate, and reintegrate incarcerated individuals [59]. Social work is an educational and practice-based profession that seeks to facilitate the welfare of communities, groups, families, and individuals. Social work, as one of the disciplines responsible for the rehabilitation of incarcerated individuals in the correctional environment, must define itself to fit well in carrying out the mandate given to the profession to make life meaningful for prisoners. The profession must also define its role appropriately to introduce and employ a consistent and accurate intervention in the rehabilitation of prisoners [59].

Correctional education has implications for social work practitioners, researchers and scholars who are interested in the well-being of incarcerated individuals. Findings revealed incarcerated individuals' perception of correctional education in Nigerian prisons. Correctional education is indeed useful to incarcerated individuals but despite its usefulness, the quality of education offered to the individuals has an influence on their participation in the programme. The quality as perceived by incarcerated individuals was seen through some factors that pose challenges to participation in correctional education. These challenges include the type of education programme, the reality of being employed after a jail sentence, security issues, inadequate learning tools, some education programmes like Internet Communication and Technology (ICT) not provided for incarcerated individuals, inconducive learning environment, language barrier, irregular class attendance by some educators, transfer of individuals from one prison to the other and individuals not allowed to take a decision on the type of

education programme to participate in. These gaps have so far affected the effectiveness of correctional education. The relevance of social workers in addressing these gaps cannot be argued. They help to rehabilitate incarcerated individuals, counsel, teach life lessons and advocate for better educational policies for incarcerated individuals [35, 51, 52].

[60] defines social work as an academic discipline that promotes social reform and growth, social stability, social harmony, equality of the poor, and liberation of people. The fundamental tenets of social work are ideals of social justice, human dignity, mutual accountability, and appreciation for diversity. This implies that the social work profession among other roles performs the function of empowering people including those in correctional institutions through education that will eventually liberate them from crime. The process correctional social workers engage in includes focusing on individual improvement through education and skill learning that will help prisoners obtain gainful employment after their release and reduce recidivism and working on processes to smooth the eventual transition from the prison back into regular society [61].

## Limitations of the study

Despite the issues raised in the study regarding education in Nigerian correctional institutions, there are limitations worth noting that should be addressed in future research. First, the study was conducted with just one prison. This is not a true representation of many incarcerated individuals in other Nigerian correctional centres. Second, the female incarcerated individuals were not adequately represented in this study. There is a need to seek the perception of female incarcerated individuals in Nigerian correctional centres. Third, the research questions are too broad and not all suitable for answering with interview data on 20 individuals. Future studies should formulate specific research questions to create room for more in-depth analysis of the interview data.

## Conclusion and recommendations

The study investigates incarcerated individuals and education programmes in Nigeria: A task for social workers. Correctional education as a means of rehabilitating incarcerated individuals is available in Nigerian prisons, especially in Owerri prison. It is seen as a very essential tool for the effective reintegration of incarcerated individuals. To enhance the participation of incarcerated individuals in education programmes, there is a need for quality education in correctional centres. The provision of quality education for incarcerated individuals is a task not only for the government and significant others but also for social workers. Social work as a helping profession has numerous roles to play. It advocates for better policies for incarcerated individuals.

Consequently, the study recommends that the government should provide quality education programmes for incarcerated individuals. This will enable them to make choices on aspects of an education programme to participate in. Employment should be given to incarcerated individuals on discharge who proved themselves worthy of a particular job after learning some skills through education while in prison. This will bring about empowerment and help in attaining the goal of correctional education. There should be effective education programmes specially designed for female incarcerated individuals. This will enhance their participation in educational programmes. Furthermore, social work is relatively a new profession in Nigeria. There is a need for awareness creation about social workers and their services. The Nigerian government as a matter of urgency should legalize the social work profession to give them the legal backing needed for practice as is the case with other countries.

## Author Contributions

**Conceptualization:** Ijeoma B. Uche, Okala A. Uche, Uzoma O. Okoye, Blessing Onyinye Ukoha-Kalu.

**Data curation:** Ijeoma B. Uche, Blessing Onyinye Ukoha-Kalu.

**Formal analysis:** Ijeoma B. Uche, Okala A. Uche, Blessing Onyinye Ukoha-Kalu.

**Investigation:** Ijeoma B. Uche, Okala A. Uche, Uzoma O. Okoye, Blessing Onyinye Ukoha-Kalu.

**Methodology:** Ijeoma B. Uche, Okala A. Uche.

**Project administration:** Ijeoma B. Uche.

**Supervision:** Uzoma O. Okoye.

**Writing – original draft:** Ijeoma B. Uche, Okala A. Uche, Blessing Onyinye Ukoha-Kalu.

**Writing – review & editing:** Ijeoma B. Uche, Okala A. Uche, Uzoma O. Okoye, Blessing Onyinye Ukoha-Kalu.

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
