## [Decision Letter · Decision Letter 0]

13 Feb 2023

PONE-D-22-29310Prison inmates and education programs in Nigeria: A task for social workers

PLOS ONE

Dear Dr. Ukoha-kalu,

Thank you for submitting your manuscript to PLOS ONE. After careful consideration, we feel that it has merit but does not fully meet PLOS ONE’s publication criteria as it currently stands. Therefore, we invite you to submit a revised version of the manuscript that addresses the points raised during the review process.

The manuscript has been evaluated by one reviewer, and their comments are available below.

The reviewer has raised concerns regarding the reporting and methodology of this study. 

Could you please revise the manuscript to carefully address the concerns raised?

Please note that we have only been able to secure a single reviewer to assess your manuscript. We are issuing a decision on your manuscript at this point to prevent further delays in the evaluation of your manuscript. Please be aware that the editor who handles your revised manuscript might find it necessary to invite additional reviewers to assess this work once the revised manuscript is submitted. However, we will aim to proceed on the basis of this single review if possible.

We look forward to receiving your revised manuscript.

Kind regards,

Johannes Stortz, PhD

Staff Editor

PLOS ONE

Journal Requirements:

2. Please provide additional information regarding the considerations  made for the prisoners included in this study. For instance, please discuss whether participants were able to opt out of the study and whether individuals who did not participate receive the same treatment offered to participants.

Reviewers' comments:

Reviewer's Responses to Questions

**Comments to the Author**

1. Is the manuscript technically sound, and do the data support the conclusions?

Reviewer #1: No

2. Has the statistical analysis been performed appropriately and rigorously? 

Reviewer #1: No

3. Have the authors made all data underlying the findings in their manuscript fully available?

Reviewer #1: Yes

4. Is the manuscript presented in an intelligible fashion and written in standard English?

Reviewer #1: Yes

5. Review Comments to the Author

Reviewer #1: The authors examine the perceptions and experiences of incarcerated individuals in Nigerian prison about education programs. The study context is really interesting because, as the authors mention, relatively little prison research comes from this part of the world. I also appreciate the data collection and analysis as described in the method section and think there are interesting findings to be published.

However, I think there is currently a mismatch between research questions, methods used and conclusions. The research questions are too broad and not all suitable for answering with interview data on 20 individuals. The authors formulate the questions as follows: The objectives of the study include 1) Education programs available for inmates. (2) Inmates’ perception and essence of correctional education. (3) The quality of correctional education programmes and inmates’ participation. (4) Social workers and correctional education.

I think the manuscript would benefit from formulating one or two more specific questions. This would also enable/create room for a more in-depth analysis of the interview data.

Another recommendation would be to more critical of the effects of educational programs and the effects of social work in a correctional setting. The authors mention this sometimes but the conclusions are still very positive about the expected effects of educational programs- even though this question is not examined in the current study and prior research shows very mixed effects.

In general, the authors draw several conclusions that do not directly follow from their study. For example, the statement ‘Employment should be given to ex-inmates…’ requires further explanation.

Minor comments:

- In recent years it has become more common to use terms as incarcerated individuals instead of inmates (thereby acknowledging the individual is more than his/her crime).

6. PLOS authors have the option to publish the peer review history of their article (what does this mean?). If published, this will include your full peer review and any attached files.

Reviewer #1: No

---

## [Author Response · Author response to Decision Letter 0]

16 Feb 2023

Thank you very much. We have uploaded a file named 'A point-by-point response to reviewers comment

---

## [Decision Letter · Decision Letter 1]

19 Jun 2023

PONE-D-22-29310R1Incarcerated individuals and education programmes in Nigeria: A task for social workersPLOS ONE

Dear Dr. Ukoha-kalu,

Thank you for submitting your manuscript to PLOS ONE. After careful consideration, we feel that it has merit but does not fully meet PLOS ONE’s publication criteria as it currently stands. Therefore, we invite you to submit a revised version of the manuscript that addresses the points raised during the review process.

We look forward to receiving your revised manuscript.

Kind regards,

Adetayo Olorunlana, Ph.D.

Academic Editor

PLOS ONE

Journal Requirements:

Reviewers' comments:

Reviewer's Responses to Questions

**Comments to the Author**

1. If the authors have adequately addressed your comments raised in a previous round of review and you feel that this manuscript is now acceptable for publication, you may indicate that here to bypass the “Comments to the Author” section, enter your conflict of interest statement in the “Confidential to Editor” section, and submit your "Accept" recommendation.

Reviewer #2: (No Response)

2. Is the manuscript technically sound, and do the data support the conclusions?

Reviewer #2: Partly

3. Has the statistical analysis been performed appropriately and rigorously? 

Reviewer #2: N/A

4. Have the authors made all data underlying the findings in their manuscript fully available?

Reviewer #2: No

5. Is the manuscript presented in an intelligible fashion and written in standard English?

Reviewer #2: Yes

6. Review Comments to the Author

Reviewer #2: The manuscript presents the importance of correctional education for incarcerated individuals, barriers to education programs, and the lack of social workers’ knowledge about correctional education. Their background in correctional education was described clearly. However, their goals and research questions were not described clearly. Therefore, the description of the method and results is insufficient.

Weakness:

1. In both the introduction and results, the authors offered the same frameworks as the objectives of the study and four themes. However, the reviewer thinks that the authors should propose more specific research questions in the introduction to assist readers to understand the hypothesis of this research. The reviewer understood that the authors recognized this issue as a limitation of research. However, the reviewer strongly advises the authors to focus on research questions connected to the research results to make it more established as a research paper.

2. In methods, the authors should provide exact information on four questions for readers to understand what kind of questions were asked. It is important to explain the relationships between the four questions and the four themes of results.

3. In the results, the authors should include the genders of the participants.

4. In Table 1, the authors should double-check if personal information is identifiable. To protect personal information, the authors could provide statistical information instead of individual data.

5. In terms of female incarcerated individuals, the authors stressed the advantage of Owerri prison including females in the introduction, but not much discussed focusing on genders. The reviewer understood the limitation of the research due to the few female candidates, but the authors could propose some discussion, depending on the female interviewees. The reviewer advises the authors to provide some ideas or recommendations from the viewpoint of gender for future practices.

Some minor points

1. Some remain as inmates. Please correct all inmates to incarcerated individuals.

2. In the sampling procedure, the authors insisted that they recruited older individuals 3 years of service and above, but in the Table 1, some participants served less than 3 years. The authors should double-check the condition of sampling.

7. PLOS authors have the option to publish the peer review history of their article (what does this mean?). If published, this will include your full peer review and any attached files.

Reviewer #2: No

---

## [Author Response · Author response to Decision Letter 1]

27 Jun 2023

Thank you for reviewing our manuscript. We have now uploaded a "point-by-by response" to the Reviewer's comment.

---

## [Editor Report · Decision Letter 2]

4 Jul 2023

Incarcerated individuals and education programmes in Nigeria: A task for social workers

PONE-D-22-29310R2

Dear Dr. Ukoha-kalu,

We’re pleased to inform you that your manuscript has been judged scientifically suitable for publication and will be formally accepted for publication once it meets all outstanding technical requirements.

Kind regards,

Adetayo Olorunlana, Ph.D.

Academic Editor

PLOS ONE
---

## [Editor Report · Acceptance letter]

12 Jul 2023

PONE-D-22-29310R2 

Incarcerated individuals and education programmes in Nigeria: A task for social workers 

Dear Dr. Ukoha-kalu:

I'm pleased to inform you that your manuscript has been deemed suitable for publication in PLOS ONE. Congratulations! Your manuscript is now with our production department. 

Kind regards, 

on behalf of

Associate Professor Adetayo Olorunlana 

Academic Editor

PLOS ONE